# Melatonin Role in Plant Growth and Physiology under Abiotic Stress

**DOI:** 10.3390/ijms24108759

**Published:** 2023-05-15

**Authors:** Irshad Ahmad, Guanglong Zhu, Guisheng Zhou, Jiao Liu, Muhammad Usama Younas, Yiming Zhu

**Affiliations:** 1Joint International Research Laboratory of Agriculture and Agri-Product Safety of the Ministry of Education of China, Yangzhou University, Yangzhou 225009, China; irshadgadoon737@yahoo.com (I.A.); g.zhu@yzu.edu.cn (G.Z.); jiaoliu0407@163.com (J.L.); 18136059863@163.com (Y.Z.); 2Key Lab of Crop Genetics & Physiology of Jiangsu Province, Yangzhou University, Yangzhou 225009, China; 3Department of Crop Genetics and Breeding, College of Agriculture, Yangzhou University, Yangzhou 225009, China; usamaghias7@gmail.com

**Keywords:** phyto-melatonin, abiotic stress, morpho-physiological activity, redox status, signaling and transduction pathway

## Abstract

Phyto-melatonin improves crop yield by mitigating the negative effects of abiotic stresses on plant growth. Numerous studies are currently being conducted to investigate the significant performance of melatonin in crops in regulating agricultural growth and productivity. However, a comprehensive review of the pivotal performance of phyto-melatonin in regulating plant morpho-physiological and biochemical activities under abiotic stresses needs to be clarified. This review focused on the research on morpho-physiological activities, plant growth regulation, redox status, and signal transduction in plants under abiotic stresses. Furthermore, it also highlighted the role of phyto-melatonin in plant defense systems and as biostimulants under abiotic stress conditions. The study revealed that phyto-melatonin enhances some leaf senescence proteins, and that protein further interacts with the plant’s photosynthesis activity, macromolecules, and changes in redox and response to abiotic stress. Our goal is to thoroughly evaluate phyto-melatonin performance under abiotic stress, which will help us better understand the mechanism by which phyto-melatonin regulates crop growth and yield.

## 1. Introduction

Abiotic stress continuously reduces crop growth and yield in different crops [1,2]. Drought, heat, salt, and heavy metals stresses are commonly produced when crops are cultivated in altered environments [1]. Crops grown in altered environments can affect physiological, biochemical, and molecular processes at various growth stages, which could lead to greater yield loss [3]. These stresses cause numerous changes in crops, including plant metabolism, hemostasis, ion distribution, and antioxidant activity. These changes have been shown to negatively affect about 70% of the yield of staple food crops [4,5]. The increase in crop yields requires better adaptation of plants to abiotic stresses [6]. Over the last few decades, research scientists have made tremendous efforts to improve crop growth and yield through the higher application of phyto-melatonin exogenous hormone.

Phyto-melatonin (*N*-acetyl-5-methoxy tryptamine) was discovered in 1995 in vascular crops [7]. Endogenously, phyto-melatonin is present in various parts of the crops, such as seeds, roots, shoots, leaves, and fruits. Phyto-melatonin enhances morpho-physiological activities, including seed germination, enhanced root enlargement, and photosynthetic activity. Phyto-melatonin applied exogenously reduces chlorophyll degradation and increases protein-related gene expression during chlorophyll synthesis [8]. Melatonin also reduces the excessive production of toxic reactive oxygen species (ROS) in animal cells [9]. To control oxidative damage’s adverse effect, melatonin is an antioxidant and free radical scavenger in early photosynthetic prokaryotic organisms [10].

Phyto-melatonin is a powerful antioxidant and can enhance crop tolerance to abiotic stress via signal regulation. During cadmium stress, phyto-melatonin efficiently modified the root growth pattern and promoted the nutrient contents of crops [11]. In addition, phyto-melatonin enhances the contents of sucrose, proline, and polyamine under cold stress [12]. In tomatoes, exogenous phyto-melatonin enhances chlorophyll contents, increases gas exchange, promotes rubisco enzyme activity, and upregulates the expression of photosynthetic-related genes under heat stress [13]. Li et al. [14] showed that phyto-melatonin mitigates the adverse effects of oxidative stress on nucleic acids, proteins, and lipids via regulating enzymatic and non-enzymatic antioxidant activities or its role as a ROS scavenger. Numerous studies demonstrated that plants treated with phyto-melatonin could mitigate the adverse effects of abiotic stress and are thus suitable for agricultural productivity. However, the role of phyto-melatonin in improving these processes under abiotic stress in different crops is still unknown. Therefore, the main purpose of this review is to study the role of phyto-melatonin in response to plant growth regulation, redox status, and its signaling and transduction pathway under various abiotic stresses.

## 2. Effects of Phyto-Melatonin under Abiotic Stresses

### 2.1. Effect on Phyto-Meltonin on Crops Growth and Yield under Abiotic Stress

Crops are continuously exposed to different abiotic stresses such as cold, salt, heavy metal, water logging, heat and ultraviolet radiation (Table 1). In fact, these stresses decreased the normal plant growth and development and as a result there were higher losses in crop yield [15,16]. Different techniques have been currently investigated to mitigate the adverse effects of abiotic stress to improve the growth and productivity of various crop plants (Figure 1). These strategies include the supply of exogenous hormones such as phyto-melatonin, rapid development in genetically modified crops, suitable cultivars, modern technologies used for modifying genes via CRISPER-Cas9, etc. [1,17]. These strategies have considerably contributed to the current agriculture systems. Nevertheless, they also have exerted some negative effects on the natural ecosystem. Thus, it is critically important to thoroughly investigate the detailed mechanisms of these strategies in order to protect the natural ecosystem from increasing harm.

Phyto-melatonin reduces the adverse effects of abiotic stresses and plays a vital role in plant growth and yield. The involvement of phyto-melatonin in regulating plant growth and yield was reported by Hernández-Ruiz et al. [18] who demonstrated that phyto-melatonin increased the coleoptiles (10–55%) of monocots, such as wheat, oat, and barley. Later, it was found in transgenic rice plants that applied phyto-melatonin promotes the initial root length, growth, and root biomass and increases seedling biomass and results in increased yield [19]. Earlier studies demonstrated that exogenous application of phyto-melatonin can mitigate the negative effects of drought stress in *Moringa oleifera* [20], salt stress in watermelon [21], and cold stress in cucumber [22]. The growth attributes and yield improvement might result from phyto-melatonin improving cell elongations, shoot water contents, reduced osmotic stress, and enhanced plant antioxidant activities. The current investigations are in unison with the studies of [23,24], who demonstrated that phyto-melatonin treatment could mitigate the adverse effects of abiotic stress and improve plant growth and yield. The present review showed that plant growth attributes and yield were both increased by phyto-melatonin under abiotic stresses. Nevertheless, the mechanism by which phyto-melatonin achieves these outcomes in various crops is still poorly understood.

**Table 1 ijms-24-08759-t001:** The performance of phyto-melatoin in different crops under abiotic stress.

Phytohormone	Crops	Abiotic Stress	Reference
Phyto-melatonin	*Moringa oleifera*	Enhance drought tolerance	[20]
	Water melon	Enahnce salt tolerance	[21]
	Cucumber	Enhance cold tolerance	[22]
	citrus	Enhance drought tolerance	[25]
	*Phaseolus vulgaris *	Enhance drought tolerance	[26]
	Soyabean	Enhance drought tolerance	[27]
	Cucumber	Enhance cold tolerance	[28]
	*Glycine max *	Enhance oxidative tolerance	[29]
	*Cucumis sativus *	Enhance oxidative tolerance	[27]
	*Vitis vinifera *	Enhance oxidative tolerance	[30]
	Cucumber	Enhance Cu tolerance	[31]

### 2.2. Effects of Phyto-Melatonin on Plant Physiological and Biochemical Activities under Abiotic Stress

Abiotic stress negatively affects plant physiological and biochemical activities in different crops [32] (Figure 2). Plants developed a comprehensive resistance system to cope with the adverse effect of these abiotic stresses [1]. The defense system of the plant contains catalase (CAT), peroxidase (POD), and superoxide dismutase (SOD) enzymes that reduce the oxidative damage caused by the extra accumulation of ROS [33]. Previous studies demonstrated that barley and wheat are exposed to salt stress and that may be due to the lower stomatal conductance or higher accumulation of ROS that enhances oxygen-induced cellular damage in plants [34,35]. Different studies demonstrated that phyto-melatonin can increase antioxidant activity and stomatal conductance in different crops [36]. The phyto-melatonion enhances plant antioxidant activities and removes the excessive production of ROS and prevents the plants from oxidative damage caused by abiotic stress [37,38]. The application of phyto-melatonin reduced drought stress and improve its morpho-physiological activities in citrus and *Phaseolus vulgaris* plants [25,26]. The phyto-melatonin enhances the plant antioxidant defense system and enhances stomatal conductance in different crops. However, the detailed mechanism of how phyto-melatonin improves the antioxidant defense system and enhances stomatal conductance in various crops under abiotic stresses is still unknown.

### 2.3. Effect of Phyto-Melatonin on Plant Redox Status under Abiotic Stress

Redox or ROS-mediated signaling has recently become a significant signaling pathway that interacts with hormone-mediated signaling [39]. Redox is a wide concept that is frequently defined as an integrated ratio of several redox couples existing inside the crop cell [40] (Figure 3). The current investigation demonstrated that each growth stage has its own redox pattern, which is regulated by the coordinated activity of several ROS enzymes such as SOD, CAT, POD, ascorbate oxidase (AO), glutathione peroxidase (GPX), and glutathione reductase (GR) [41].

Abiotic stress resulted in the production of redox status in plants [42]. Despite ROS functioning as signaling molecules at low levels, the excessive accumulation of ROS causes oxidative stress and reduces plant growth and damage to macromolecules [43]. Thus, it is critically important to understand how stress influences the redox status in plant growth and yield.

Previous studies demonstrated that phyto-melatonin reduces the higher accumulation of ROS and reactive nitrogen species (RNS) and enhances the cell’s redox status under abiotic stress [44,45]. The ROS and RNS can be regulated through melatonin-mediated stimulating redox enzymes, including SOD, POD, CAT, and ascorbate peroxidase (APX) [46]. The phyto-melatonin transportation takes place through the xylem to other plant parts. Some studies believe that phyto-melatonin transportation takes place in chloroplast and mitochondria [47].

The phyto-melatonin reduces heat stress in tomatoes and soybeans through balanced redox via nitric oxide modulation and polyamine biosynthesis [28,48]. In cotton, phyto-melatonin treatment regulates the expression of redox-related genes and mitigates the adverse effects of salt stress [2]. The phyto-melatonin application modulates some leaf senescence proteins, which further interact with photosynthesis regulation and macromolecules and, as a result, affects redox changes and response to abiotic stresses in plants [30,49]. Nevertheless, various proteins involved in changing redox status and response to multiple stresses still need to be clarified. In addition, phyto-melatonin upregulates the expression of redox-related genes to mitigate abiotic stress, which requires further investigation.

### 2.4. Effect of Phyto-Melatonin in Plant Defense Systems under Abiotic Stress

Due to changes in climate, abiotic stress disturbs plants’ morpho-physiological activities and yield [1,50,51]. Abiotic stress alters the functioning of several physiological processes and overall metabolic health by substantially increasing the formation of ROS, resulting in damage to lipids and proteins and, in extreme cases, plant cell death. To mitigate the adverse effects of ROS, plants possess a variety of defensive responses such as SOD, CAT, and POD to deal with abiotic stresses [43,52]. However, when plants are exposed to these stresses longer, the defensive mechanisms fail to protect them against ROS damage [53]. Plants have other various resistance pathways to cope with the adverse effects of ROS damage in plants caused by abiotic stress, in which the application of exogenous hormones is, in fact, significant.

Phyto-melatonin confers several benefits in plants such as mitigating the adverse effects of abiotic stresses and improving the plant defense system [1]. Applying phyto-melatonin improves plant morpho-physiological and antioxidant activities such as SOD, POD, and CAT in soybean and cucumber under drought stress [27,28]. Similarly, phyto-melatonin improves plant antioxidant activities and prevents the plant from chromium stress due to the higher production of phyto-chelatains and compartmentalization of chromium in the vacuole and cell wall. Phyto-melatonin enhances the plant defense system via higher production of phyto-chelatains [46]. Different strategies, such as macro-nutrients and genes [31], showed that phyto-melatonin improves the plant defense system in various crops under abiotic stress. However, the detailed mechanism still needs to be further investigated.

### 2.5. Phyto-Melatonin Signaling and Transduction Pathway under Abiotic Stress

Crops use various mechanisms to respond to environmental changes. A good example is the opening and closing of stomata, which are regulated by phyto-melatonin hormonal signals [54]. The signaling and transduction pathways of phyto-melatonin are largely unknown due to a lack of knowledge (Figure 4). Stomata regulate the uptake of CO_2_ and release water vapor, which is necessary for plant growth and yield [55]. In *Arabidopsis*, the receptor CAND2/PMTR1-dependent phyto-melatonin signaling regulates the opening and closing of stomata via G-protein α subunit-regulated H_2_O_2_ and Ca^2+^ signals transduction cascades [56]. Phyto-melatonin is a new phyto-hormone that controls the opening and closing of stomata in different plants. However, the role of the phyto-melatonin in CAND2/PMRT1-dependent receptors in the opening and closing of stomata under abiotic stress is still unknown.

Moreover, melatonin modulates the gene expression level in the jasmonic, salicylic, and ethylene pathways [47]. Melatonin upregulated ethylene signals transduction elements such as the ERF2, EIL1, and EIL3 and receptor genes NR and ETR4 in tomatoes and enhanced its production during fruit ripening [49]. The phyto-melatonin-treated plants, such as carrots and cucumbers, showed a higher level of putrescine and spermidines in plant cells [57]. Previous studies showed that the actions of phyto-melatonin may also involve other phytohormones [58]. Putrescine and spermidine both play a crucial role in plant cell growth. However, the mechanism of how melatonin increased the levels of spermidine and putrescine under abiotic stress in different crops still needs to be clarified.

### 2.6. Phyto-Melatonin Acts as a Stimulator under Abiotic Stress

Biostimulants, plant hormones, enzymes, vitamins, proteins, amino acids, micronutrients, and other compounds play a vital role in modifying and regulating physiological processes in plants to promote plant growth and nutrient uptake, alleviate the adverse effects of abiotic stress, and increase yield. Abiotic stress prevents crops from achieving higher yields, but plant growth regulators, including phyto-melatonin, gibberellin, cytokinin, brassinosteroids, auxin, and mepiquate chloride, may reduce these abiotic stresses.

The stimulatory effect of exogenous phyto-melatonin has been widely investigated in different crop studies. It has been demonstrated to be a signal molecule that regulates several physiological and biochemical activities [59,60]. Further investigations showed that phyto-melatonin as a biostimulant enhances plant growth, fruit shelf life, fruit quality, tolerance to abiotic stress, and yield in different plants [61,62,63]. Phyto-melatonin, as a biostimulant, reduces the adverse effects of different kinds of abiotic stresses in various plants [17]. In addition, several studies demonstrated that melatonin as a biostimulant reduced the negative effects of oxidative stress induced by water deficit for *Glycine max* [29], *Cucumis sativus* [27], and *Vitis vinifera* [30]. It has been suggested that phyto-melatonin has the highest antioxidant capacity among all plant growth regulators and is recognized as a biological stimulant with very intense antioxidant activities [64]. In cauliflower and red cabbage, applying phyto-melatonin improves the plant’s seedling and vegetative capacity under higher Cu concentrations [65,66].

Similarly, in cucumber, the phyto-melatonin mitigates the adverse effects of Cu (copper) stress on plants due to Cu sequestration, ROS, and carbon metabolism [31]. Phyto-melatonin acts as a biostimulant, reduces the negative effects of different kinds of abiotic stress, and enhances various plants’ morpho-physiological activities and yield.

## 3. Role of Phyto-Melatonin in Crops under Abiotic Stress

### 3.1. Cold Stress

Cold stress reduces plant morpho-physiological, biochemical, molecular, and metabolic activities in various crops and, as a result, it reduces crop growth and yield [1,67]. Cold stress occurs between 0 °C and 12 °C [68]. Cold stress significantly decreases a plant’s membrane fluidity, antioxidant enzymes, and metabolism homeostasis [69,70]. Cold stress also reduces the crop’s photosynthesis activities by disrupting the carbon–oxygen cycle, CO_2_ supply, and the production of photosynthesis pigments [71]. Photosynthesis is highly susceptible to cold stress as low temperatures hinder the major components of plants’ photosynthetic activities [72]. During cold stress, reducing chlorophyll content leads to chlorosis in leaves [73]. The chlorophyll contents of leaves deliver important information about the effectiveness of physiological processes in plants [74]. Previous studies showed that plants treated with phyto-melatonin had a higher chlorophyll content than non-treated plants under cold stress [7]. Growing plants under cold stress leads to oxidative stress via increased ROS, such as superoxide anion, hydroxyl radicals, and hydrogen peroxide [69,75]. During cold stress, the higher accumulation of ROS significantly affects DNA structures, lipids peroxidation, and protein synthesis oxidation within plant cells, hindering plant growth and yield [76,77]. To cope with the adverse effects of oxidative stress caused by ROS, plants develop a defensive response, including non-enzymatic antioxidants such as proline and glutathione and enzymatic antioxidants such as SOD, CAT, and POD [78,79]. Different studies showed that phyto-melatonin enhanced plant growth and yield by eliminating the excessive production of ROS during cold stress [80]. Phyto-melatonin can stimulate the morpho-physiological activities of various crops, including corn, and promote the germination of cucumber under cold stress. In rice, phyto-melatonin reduces cold stress by regulating antioxidant activities and cold-related genes [81]. Seeds treated with phyto-melatonin promote the yield of early planted spring maize by stimulating early seedling growth under cold stress [3]. Altaf et al. [82] demonstrated that phyto-melatonin reduced the adverse effects of cold stress in pepper by recovering root traits, plant growth, exchange of gases, pigment molecules, and related gene expression as compared to the non-cold stress plants. Phyto-melatonin increases the crop’s growth and yield by improving its morpho-physiological and molecular activities under cold stress. However, the underlying mechanism of phyto-melatonin in different crops under cold stress is still lacking.

### 3.2. Salt Stress

Salt stress negatively affects plant growth and yield. The exogenous application of phyto-melatonin mitigates the adverse effects of salt stress and improves plant morpho-physiological, biochemical, and molecular activities [36]. In the current research, more effort has been focused on the role of phyto-melatonin in mitigating salt stress in plants. However, information regarding phyto-melatonin in regulating seed germination and phyto-melatonin-related genes under salt stress is still lacking [83,84].

It has been confirmed that seed germination is an important stage during plant growth because it is a critical stage that determines the establishment of the next generation of plants. Previous studies investigated that seeds treated with phyto-melatonin had a higher germination rate and higher antioxidant activities than the control [36]. During abiotic stress, phyto-melatonin enhances the germination rate of *Zizyphus* due to the generation of antioxidant activities and the involvement of osmolytes in scavenging ROS [64]. Similarly, in cotton, phyto-melatonin-treated plants improve seed germination by modulating the expression of phyto-melatonin signaling genes, thus resulting in a varied metabolic process and the reduction of cotton seedlings under salt stress [85]. It has been investigated that phyto-melatonin reduces ABA and increases GA contents during the early germination stage compared to NaCl treatment [84]. It has been confirmed that exogenous phyto-melatonin can reduce the adverse effects of NaCl during seed germination. However, the performance of phyto-melatonin during various growth stages under salt stress is still in an early phase.

Previous studies demonstrated that melatonin promotes ethylene biosynthesis, and the genes *ACS1* and *MYB108A* were highly induced by phyto-melatonin treatment during grapevine ripening under salt stress [86]. Moreover, phyto-melatonin-related genes such as *COMT* reduce the adverse effects of salt stress and improve crop growth and yield [21]. The *COMT* genes have been widely identified in plants such as *Populous tremuloides*, *Hordeum vulgare*, *Chrysanthemum grandiflorum*, and *Lolium perenne* [87,88,89]. The involvement of *COMT* in phyto-melatonin biosynthesis was only investigated in model plants such as monocotyledonous, including rice, and dicotyledonous, and including *Solanum lycopersicum*, and *Arabidopsis*.

### 3.3. Heavy Metal Stress

Heavy metals such as iron (Fe), copper (Cu), molybdenum (Mo), and nickel (Ni) are required for plant growth and metabolism and their higher accumulation can adversely affect plant growth and yield [1]. While arsenic (As), cadmium (Cd), and lead (Pb) are highly harmful to plant growth and yield and not required by plants [89]. Heavy metals hinder many physiological and biochemical activities, such as photosynthesis and antioxidant activities in carbohydrate metabolism and the Calvin cycle [90]. During heavy metal stress, the higher accumulation of ROS inhibits root growth and promotes leaf senescence [91]. Different studies showed that the exogenous application of phyto-melatonin improves plant growth and yield by enhancing its physiological and biochemical activities in various crops [1]. Hoque et al. [92] showed that applying phyto-melatonin eliminates the negative effects of heavy metal stress by improving the plant’s nutrient balance, redox status, and osmotic regulation. Applying phyto-melatonin reduces heavy metal stress, improves seed germination in cucumber and red cabbage [65], and improves soybean antioxidant activities [83]. Melatonin protects plants from the translocation of heavy metal stress and upregulates the genes involved in phyto-melatonin biosynthetic pathways [93]. For example, applying phyto-melatonin upregulates the overexpression of *HSfA1a* gene, which enhances plant tolerance to Cd stress in tomato [94]. Plant growth and yield were enhanced under heavy metal stress via the upregulation of melatonin biosynthetic genes. However, the mechanism by which various genes are involved in the biosynthetic pathway of phyto-melatonin under heavy metal stress still needs to be improved in different crops [1].

### 3.4. Ultraviolet Stress

Ultraviolet (UV) radiation continuously increases, caused by the rapid ozone layer depletion, negatively affecting plant growth and yield [1]. The increase in UV radiation can substantially reduce crop productivity by hindering plants’ morpho-physiological activities [15]. Higher UV radiation levels reduced the synthesis of key photosynthesis proteins, i.e., chlorophyll *a/b* binding proteins [90]. Phyto-melatonin plays an important role in reducing the negative effect of UV radiation on plant growth and productivity. It has been shown that phyto-melatonin in *Arabidopsis thaliana* reduced the negative stress of UV radiation by improving crop signaling transduction pathways, i.e., transcription factors HY5, HYH, and RUP/2, and the ubiquitin-degrading enzyme COP1 [91,92]. Phyto-melatonin is considered a potent antioxidant; thus, it regulates antioxidant defense systems to protect plants from the negative impacts of UV radiation [92]. Phyto-melatonin reduced the negative effects of UV stress on plant photosynthetic parameters such as chlorophyll levels, leaf membrane damage, stomatal conductance, and gas exchanges in *Malus hupehensis* [91]. Phyto-melatonin enhances plant growth and development in various crops. However, the underlying mechanism of phyto-melatonin in regulating different signaling pathways and elevating the adverse effects of UV radiation in various crops is still lacking.

### 3.5. CO_2_ Stress

Phyto-melatonin improves the contents of proteins related to the thioredoxin-peroxiredoxin pathway and ascorbic acid-glutathione cycle in leaves exposed to NO_2_ stress, regulating their redox balance [8]. Plants absorb NO_2_ from the atmosphere to purify and metabolize the air [93]. The low concentration of NO_2_ can be used as a source of nitrogen or signal to trigger crop growth and yield [94]. Nevertheless, a higher concentration of NO_2_ can cause damage to crops. The higher accumulation of NO_2_ causes ROS and RNS and as a result toxicity occurs in plants [95]. The low levels of ROS can be used as an intracellular signaling molecule, balanced by the crop’s antioxidant defense systems. While higher levels of ROS reduce plant photosynthetic activity and carbon fixation and affect the crop metabolism network [96,97]. During NO_2_ stress, phyto-melatonin mediates the polyamine biosynthetic pathway and enhances the expression of key enzymes and proteins such as SAMS1, SAMS2, and SAMS3 in polyamine biosynthetic pathway in plants leaves [8]. The Phyto-melatonin regulates ABA signaling transduction pathways and calmodulin-binding transcription factors such as CAMTA12 and NtCaM calmodulin NtCaM2 in Ca^2+^ signal transduction. The evidence from these studies indicated that phyto-melatonin could alleviate NO_2_ stress in crops and is thus suitable for agricultural productivity. However, to take these studies into further consideration, the defensive role of phyto-melatonin in improving these processes under abiotic stresses is still unknown.

## 4. Conclusions

Abiotic stresses on plant growth are considered a significant threat to agricultural yield. Plants use different physiological, biochemical, and molecular responses to mitigate the adverse effects of abiotic stresses [1,16]. The current review demonstrated that phyto-melatonin could reduce or mitigate the adverse effects of abiotic stress in different crops. The exogenous application of phyto-melatonin increases plant growth and development and reduces the adverse impact of abiotic stresses on other plants. Phyto-melatonin is crucial in further plant growth regulation, redox status, and signal transduction under abiotic stresses. However, the role that phyto-melatonin plays in the underlying mechanisms of plants grown under abiotic stress is still unknown.

## 5. Future Recommendations

It has been confirmed that the application of melatonin modulated leaf senescence protein and that protein further interacts with photosynthesis regulation, macromolecules, and changes in redox status by which plants respond to abiotic stress. However, various proteins involved in changes in redox status to control abiotic stress are still largely unknown.

In *Arabidopsis*, the opening and closing of stomata are regulated by receptor CAND2/PMTR1-dependent phyto-melatonin signaling through Ca^2+^ flux and Gα subunit-mediated H_2_O_2_ signaling transduction cascade. However, the performance of the receptor CAND2/PMRT1-dependent phyto-melatonin signaling in regulating stomata under abiotic stress remains unknown in various crops.

It has been confirmed that plants treated with phyto-melatonin had higher putrescine and spermidine (polyamine) contents. Putrescine and spermidine play an important role in plant cell growth. However, the mechanism phyto-melatonin increases the levels of putrescine and spermidine under abiotic stress in different crops is still unknown.

## Figures and Tables

**Figure 1 ijms-24-08759-f001:**
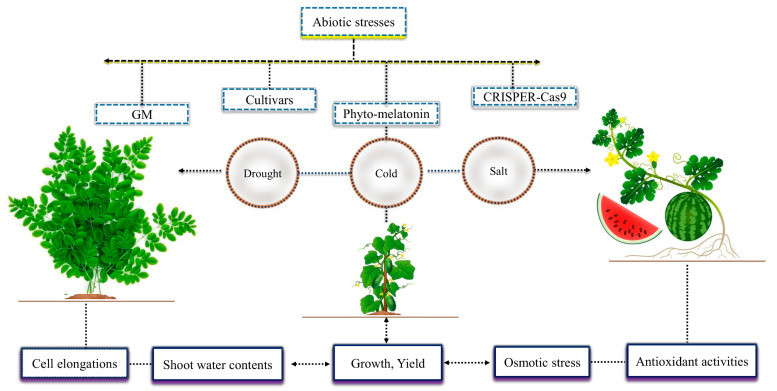
Different techniques reduce the negative effects of abiotic stresses in crops, including phyto-melatonin. The exogenous phyto-melatonin reduced the adverse effect of cold stress in cucumber, salt in watermelon, and drought in *Moringa oleifera* and improved plant growth attributes such as root length, growth and root biomass, seedling biomass and yield due to cell elongation, shoot water contents, regulated osmotic stress, and antioxidant activities.

**Figure 2 ijms-24-08759-f002:**
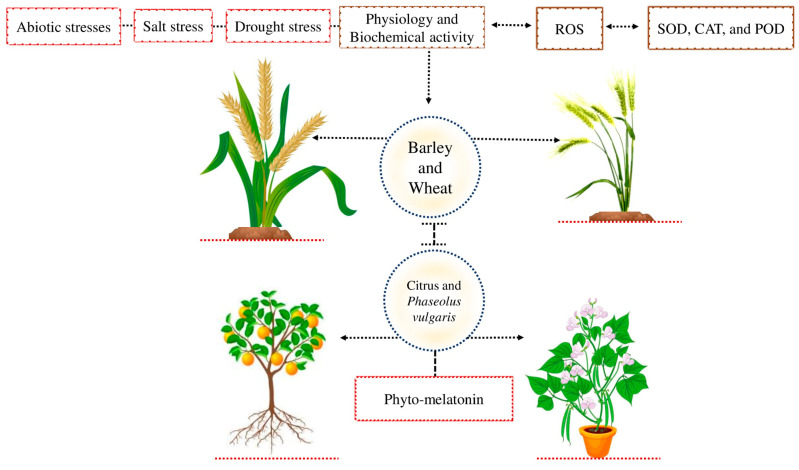
Abiotic stress reduces plant physiology and biochemical activity due to the higher accumulation of ROS. Plants produce various antioxidant activities, such as CAT, POD, and SOD, to cope with the adverse effects of ROS. Wheat and barley are exposed to salt stress due to the lower stomatal conductance or higher accumulation of ROS and, as a result, cause cellular damage. Applying photo-melatonin improves plant stomatal conductance and protects the plants from a higher accumulation of ROS in both crops. In addition, phyto-melatonin reduces the negative effects of drought stress and improves its morpho-physiological activity in citrus and *Phaseolus vulgaris*.

**Figure 3 ijms-24-08759-f003:**
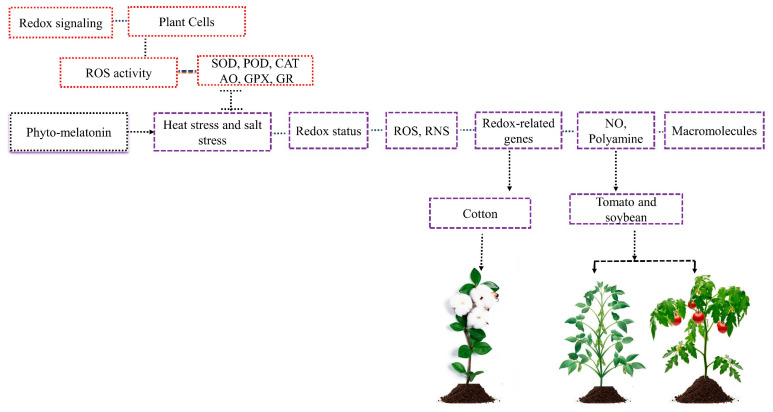
Redox signaling is present in plant cells which are regulated by plant several ROS enzymes such as CAT, POD, and SOD. Abiotic stresses increase redox status, causing higher accumulation of ROS and RNS that reduce plant growth and damage to macromolecules. The application of phyto-melatonin regulates cell redox in plants, reduces the higher accumulation of ROS and RNS, improves its growth, and prevents macromolecules from damage. The phyto-melatonin balances redox via nitric oxide and polyamine, reduces the heat stress in tomato and soybean, and enhances the salt stress in cotton via redox-related genes.

**Figure 4 ijms-24-08759-f004:**
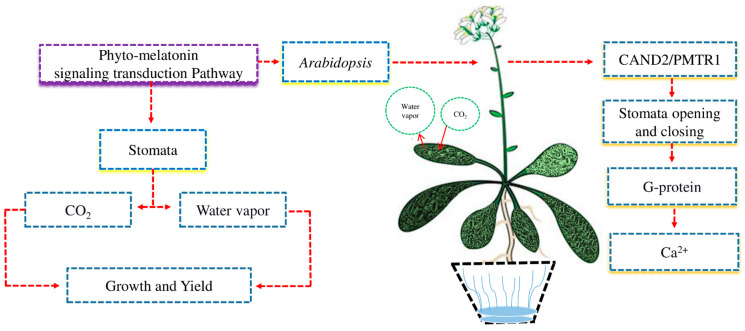
The phyto-melatonin regulates the opening and closing of stomata under abiotic stress. Stomata regulate the intake of CO_2_ and release water vapor which can enhance plant growth and yield. In *Arabidopsis*, phyto-melatonin signaling receptor CAND2/PMTR1 regulated the opening and closing of stomata via heterotrimeric G-protein to regulate Ca^2+^ flux.

## Data Availability

Not applicable.

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
