# Peer review of "Melatonin Role in Plant Growth and Physiology under Abiotic Stress"

_ijms, 2023, doi:10.3390/ijms24108759_

Round 1
Reviewer 1 Report
This review dealts with new and important agricultural problem - using compounds known as neurotransmitters of animals. A lot of publications induced of the interest to the substances, however there is no any ideas for the following moving. As a cliche of MDPI ( standard and only standard , pioneers are not received at all) it is good review and may be published here.
3. As for common journal in molecular science, author should show a link between all living organisms via common regulators- biomediators-neurotransmitters- signaling compounds.
Some questions for the improving:
1. Why do you use term " phytomelatonin? What is the difference in the structure or this is the same one that is found in animals? Principial difference?
2. It would be better if the authors may compare functions of melatonin in animals and in plants in short phrases.
Author Response
Response to Reviewer 1
Comments and Suggestions for Authors
This review dealts with new and important agricultural problem - using compounds known as neurotransmitters of animals. A lot of publications induced of the interest to the substances, however there is no any ideas for the following moving. As a cliche of MDPI (standard and only standard, pioneers are not received at all) it is good review and may be published here.
- As for common journal in molecular science, author should show a link between all living organisms via common regulators- biomediators-neurotransmitters- signaling compounds.
Ans. Respected reviewer, thank you very much for improving the work. The review only focuses on the role of phyto-melatonin in plant growth and yield under abiotic stress. The link between all living organisms via common regulators-mediator-neurotransmitters-signaling compounds is a separate hot topic in current research and may be suitable for another review paper.
Some questions for the improving:
- Why do you use term " phytomelatonin? What is the difference in the structure or this is the same one that is found in animals? Principial difference?
Answer. The term Phyto-melatonin “refers to melatonin of plant origin.” Melatonin was first observed in animals, and later, after a long observation in 1995, it has been suggested that it is present in plants too. That’s why it’s called phyto-melatonin. There are no differences in the structure of phyto-melatonin found in animals and plants.
It would be better if the authors may compare functions of melatonin in animals and in plants in short phrases.
Ans. Thank you very much, the short comparative functions of melatonin in animals has been added to the manuscript but again the review only focusing on plants melatonin. Thank you so much.

Reviewer 2 Report
Dear Authors
I have a concern about your article to publish.
1. what is the novelty of this study?
2. why are you using self-citation a lot!!!?
Please add the conclusion part.
I did not see the line number too!!
Some parts need English revision because it is not clear.
Some parts need English revision because it is not clear. Please revise the article based on the English editing
Author Response
Response to Reviewer 2
Dear Authors
I have a concern about your article to publish.
- what is the novelty of this study?
Ans. Dear reviewer, Thank you very much. Abiotic stress currently reduces crop growth and yield worldwide. Phyto-melatonin is a new hormone that promotes the plant's morpho-physiological and biochemical activities in different crops. The detailed work on this hormone in various crops under abiotic stresses still needs to be documented. Therefore, the current review provides detailed knowledge of how phyto-melatonin enhances plant growth regulation, redox status, signal transduction, and yield in plants under different abiotic stresses in various crops.
why are you using self-citation a lot!!!?
Ans. Dear reviewer, I apologize; the work I cited has already been published, is confidential, and is suitable to cite according to the current research situation, which is beneficial for future research scientists.
- Please add the conclusion part.
Ans. The conclusion parts has been added.
- I did not see the line number too!!
Ans. Dear reviewer the line numbering has been added to the manuscript.
- Some parts need English revision because it is not clear.
Ans. thank you very much for improving our work. The paper has been revised by a native English speaker.

Reviewer 3 Report
Review
Manuscript ID: ijms-2372430
“Phyto-Melatonin Role in Plant Growth and Physiology Under Abiotic
Stress”
Irshad Ahmad, Guanglong Zhu, Guisheng Zhou *, Jiao Liu, Muhammad
Usama Younas, Zhuyi Ming
Increasing the resistance of crops to negative abiotic factors is an important problem of crop production. Deciphering the mechanisms of plant protection allows us to find ways to regulate the state of plants under stressful conditions. In this regard, the presented manuscript is relevant.
The authors studied the results of studies of a large number of scientists (97) concerning the effect of adverse abiotic factors on plants. The authors cited mostly recent publications (in the last 5 years). They showed the involvement of melatonin in mitigating the effects of different stressors and highlighted several plant defense mechanisms. The review is divided into 4 sections. There are 4 figures and 1 table in the article.
The content of the manuscript text raises many questions. Great changes should be made: change the style of the text, remove repetitive phrases throughout the text, expand the description of growth and yield traits, and clarify the figures. The sequence of sections should be reconsidered. The text and figures should show general and specific plant protection systems in response to different types of negative factors.
Locations of misinterpretation or misprints are highlighted in color. Since there is no line numbering, the remarks are attributed to the sections of the manuscript.
Title
The use of "phyto-melatonin" in the title of the manuscript has no basis, since the review contains mainly studies of exogenous applications of melatonin.
Abstract
The authors use the phrase "protein expression" in the abstract and text. However, the term "expression" is used in relation to genes: "gene expression".
1
(1) It is not clear why, in the Introduction of the manuscript, the authors address private issues of NO2-stress. General well-known facts about melatonin should be given.
(2) The style of narration should be changed in the first paragraph. The repetition of similarly constructed turns deteriorates the text.
2
2.1
(1) The description of the signs of growth and yield should be expanded.
(2) The description of the direction of action of melatonin in the text does not match the information in Table 1. In the text, phytomelatonin mitigates stress, whereas in the table it increases stress from drought and other factors.
(3) The description of the sequence of events is broken in some sentences. For example, the caption of Figure 3: the description of changes in molecules (damage to macromolecules) should precede the change in processes (growth).
(4) The authors conclude that melatonin is involved in crop growth and yield. At the same time, they cite only cell elongation as a growth process. It is not clear from section 2.1 how the growth of plant organs changes and how crop traits change. However, the authors write, "The present review showed that plant growth attributes and yield were both increased by phyto-melatonin under abiotic stresses."
(5) Figure 2 is not very clear. The authors indicate one of the abiotic stressors, drought, but do not indicate salinity, although they write about it in the text. In addition, they show barley and wheat in the figure, but do not mention citrus and beans. The different parameters are depicted with the same figures, which makes it difficult to see.
(6) Structure of the review. In my opinion, the text should first describe stress-induced damage (morphogenesis, oxidative status) and then the functioning of the defense system (enzymatic and non-enzymatic antioxidants). The antioxidant functions of melatonin should be added, specifying the intermediates and endpoints resulting from melatonin oxidation. oxidative status), and then the functioning of the protective system (enzymatic and non-enzymatic antioxidants). (Melatonin's own antioxidant functions should be added, specifying intermediates and end substances derived from the oxidation of melatonin.)
2.3
(1) Figure 3 should be clarified. The authors show both the general scheme and here the private examples, without concatenating themExamples of stress types should be attached using arrows (or otherwise) to each general item in the chart. For example, Abiotic stress (salinity, drought)....
(2) The authors write about the transport of melatonin into chloroplasts and mitochondria [44]. However, nothing is said about the place of melatonin synthesis in the cell. Perhaps we should talk about the synthesis rather than the transport of melatonin into organelles.
Indicate in the text the place of melatonin synthesis, as well as the dynamics of its content under stress factors.
2.4
1 Paragraph
(1) Let me disagree with the 1st sentence of the section. Abiotic stress is always a critical determinant of crop yield.
(2) What defensive reactions are characterized by plants? You don't name these reactions, but you go on to write that there are other different pathways of resistance.
2 Paragraph
(1) The repeated citation of reference 43 should be removed.
(2) At the end of the section you write about a detailed mechanism (what mechanism?)...
2.5
(1) Figure 4 should be signed at the bottom.
(2) Figure 4 should be modified. Phytomelatonin alters Ca2+ content through the receptor, not the other way around as shown by the authors. It is necessary to connect Ca-signaling with stomata (their movement)?
2.6
At the end of the section, you write about the possible effectiveness of other growth regulators. Why do you write about that? This fact is irrelevant to the topic of the review.
3
3.1
What important information does the chlorophyll content convey to plants? Clarify your phrase.
3.2
2 Paragraph
(1) The terminology is used incorrectly. Seed germination is not a stage of plant regeneration. This phenomenon is the first stage in the life of a new organism.
(2) Reference 83 should be corrected.
3.3
Plant responses to chromium and other heavy metals should be combined into one section (2.4 and 3.3).
3.4
Section 3.4 should be named according to the factor in question.
4
At the end of the section, you write about the possible effectiveness of other growth regulators. Why do you write about that? This fact is irrelevant to the topic of the review.
References
There are many typos in the reference list. The rudest typo: the lack of italics for the Latin names of plant species.
All figures in the text should be referenced.

Author Response
Response to Reviewer 3.
increasing the resistance of crops to negative abiotic factors is an important problem of crop production. Deciphering the mechanisms of plant protection allows us to find ways to regulate the state of plants under stressful conditions. In this regard, the presented manuscript is relevant.
The authors studied the results of studies of a large number of scientists (97) concerning the effect of adverse abiotic factors on plants. The authors cited mostly recent publications (in the last 5 years). They showed the involvement of melatonin in mitigating the effects of different stressors and highlighted several plant defense mechanisms. The review is divided into 4 sections. There are 4 figures and 1 table in the article.
Answer. Thank you so much.
The content of the manuscript text raises many questions. Great changes should be made: change the style of the text, remove repetitive phrases throughout the text, expand the description of growth and yield traits, and clarify the figures. The sequence of sections should be reconsidered. The text and figures should show general and specific plant protection systems in response to different types of negative factors.
Locations of misinterpretation or misprints are highlighted in color. Since there is no line numbering, the remarks are attributed to the sections of the manuscript.
Ans. Dear reviewer, thank you very much for improving our work, and I appreciate your precious time in revision. We have tried our best to improve our manuscript according to your comments.
Title
The use of "phyto-melatonin" in the title of the manuscript has no basis, since the review contains mainly studies of exogenous applications of melatonin.
Ans. Dear reviewer, Okay the title has been changed accordingly.
Abstract
The authors use the phrase "protein expression" in the abstract and text. However, the term "expression" is used in relation to genes: "gene expression".
Ans. Thank you very much. You are right. We made a mistake. The term expression has been removed, and the sentence has been rectified accordingly in the abstract and text.
1
- It is not clear why, in the Introductionof the manuscript, the authors address private issues of NO2-stress. General well-known facts about melatonin should be given.
Ans. The main focus of the current review paper is to point out the role of phyto-melatonin under various abiotic stresses. Therefore, in the manuscript's introduction section, I have first mentioned the source of NO2- stress and its importance and then the stress prevention by photo-melatonin but in detail. Supposedly, the low concentration of NO2 can be used as a nitrogen source or signal to enhance crop growth and productivity. But the higher accumulation of NO2 causes reactive oxygen species (ROS) that damage plants. The application of Phyto-melatonin could reduce the adverse effect of ROS and reduce the adverse effect of NO2 stress. The general knowledge of melatonin has already been discussed in the above paragraph. If I discussed it again, repeating a similar construct deteriorates the text.
- The style of narration should be changed in the first paragraph. The repetition of similarly constructed turns deteriorates the text.
Ans. The repetition has been removed accordingly thank you very much.
2
2.1
(1) The description of the signs of growth and yield should be expanded.
Ans. The description has been added to the section.
- The description of the direction of action of melatonin in the text does not match the information in Table 1. In the text, phytomelatonin mitigates stress, whereas in the table it increases stress from drought and other factors.
Ans. Thank you very much for highlighting this mistake. The issue has been resolved. Actually, in the table, It is “enhance drought tolerance etc.” not stress.
(3) The description of the sequence of events is broken in some sentences. For example, the caption of Figure 3: the description of changes in molecules (damage to macromolecules) should precede the change in processes (growth).
Ans. Figure 3 and almost all the figure has been changed according your comments.
- The authors conclude that melatonin is involved in crop growth and yield. At the same time, they cite only cell elongation as a growth process. It is not clear from section 2.1 how the growth of plant organs changes and how crop traits change. However, the authors write, "The present review showed that plant growth attributes and yield were both increased by phyto-melatonin under abiotic stresses."
Ans. The growth attributes enhanced by phyto-melatonin have been added to the sections. In addition, the figure has been changed, too, according to the comments. The new one has replaced the old figure.
- Figure 2is not very clear. The authors indicate one of the abiotic stressors, drought, but do not indicate salinity, although they write about it in the text. In addition, they show barley and wheat in the figure, but do not mention citrus and beans. The different parameters are depicted with the same figures, which makes it difficult to see.
Ans. Dear reviewer, thank you very much. The abiotic stress, such as salt stress, has been added in the right place in Figure 2. In addition, citrus and beans have also been added to the figure to clarify it. The new one has replaced the old figure.
- Structure of the review. In my opinion, the text should first describe stress-induced damage (morphogenesis, oxidative status) and then the functioning of the defense system (enzymatic and non-enzymatic antioxidants). The antioxidant functions of melatonin should be added, specifying the intermediates and endpoints resulting from melatonin oxidation. oxidative status), and then the functioning of the protective system (enzymatic and non-enzymatic antioxidants). (Melatonin's ownantioxidant functions should be added, specifying intermediates and end substances derived from the oxidation of melatonin.)
Dear reviewer, thank you very much; I think the current format of the articles is suited and follows each other’s step-by-step.
2.3
- Figure 3should be clarified. The authors show both the general scheme and here the private examples, without concatenating themExamples of stress types should be attached using arrows (or otherwise) to each general item in the chart. For example, Abiotic stress (salinity, drought).
Ans. Dear reviewer, the figure has been clarified accordingly and replaced by the old one.
- The authors write about the transport of melatonin into chloroplasts and mitochondria [44]. However, nothing is said about the place of melatonin synthesisin the cell. Perhaps we should talk about the synthesis rather than the transport of melatonin into organelles.
Indicate in the text the place of melatonin synthesis, as well as the dynamics of its content under stress factors.
Ans. Respected reviewer, you may be right. But when the hormones are exogenously applied to plants, it's important to know about the transportation of melatonin in plants. In addition, it's quite detailed research and requires a huge amount of time to study the signaling and transduction of melatonin in plants.
2.4
1 Paragraph
- Let me disagree with the 1st sentence of the section. Abiotic stress is always a critical determinant of crop yield.
Ans. Okay thank you the sentence has been revised.
- What defensive reactions are characterized by plants? You don't name these reactions, but you go on to write that there are other different pathways of resistance.
Ans. The defensive response such as SOD, CAT and POD has been added to the sentence.
2 Paragraph
- The repeated citation of reference 43 should be removed.
Ans. The repeated citation 43 has been removed from the text.
- At the end of the section you write about a detailed mechanism (what mechanism?).
Ans. The role of phyto-melatonin in plants' various defensive responses such as CAT, POD, SOD, etc., and their relative genes, the genes involved in their signaling pathway, all these, you can say.
2.5
(1) Figure 4 should be signed at the bottom.
Ans. Fig 4. Caption has been signed at the bottom of the figure.
- Figure 4 should be modified. Phytomelatonin alters Ca2+ content through the receptor, not the other way around as shown by the authors. It is necessary to connect Ca-signaling with stomata (their movement)?
Ans. Thank you very much. Yes, I understand. The figure has been modified according. The connection of Ca2+ has been linked with the receptor, and more information has been added for more clearance to the figure.
2.6
At the end of the section, you write about the possible effectiveness of other growth regulators. Why do you write about that? This fact is irrelevant to the topic of the review.
Ans. The other growth regulator has been deleted as it seems irrelevant to the topic of the review.
3
3.1
What important information does the chlorophyll content convey to plants? Clarify your phrase.
Ans. The phrase has been clarified.
3.2
2 Paragraph
- The terminology is used incorrectly. Seed germination is not a stage of plant regeneration. This phenomenon is the first stage in the life of a new organism.
Ans. Okay the modification has been done.
- Reference 83 should be corrected.
Ans. Okay thank you the reference has been corrected.
3.3
Plant responses to chromium and other heavy metals should be combined into one section (2.4 and 3.3).
Ans. The stresses have been combined, and the names have been changed to other stresses.
Section 3.4 should be named according to the factor in question.
Ans. The section 3.4 named has been changed to UV stress.
4
At the end of the section, you write about the possible effectiveness of other growth regulators. Why do you write about that? This fact is irrelevant to the topic of the review.
Ans. The information on the other growth regulator has been deleted as it seems irrelevant to the topic. Thank you.
References
There are many typos in the reference list. The rudest typo: the lack of italics for the Latin names of plant species.
Ans. All the references have been changed accordingly.
All figures in the text should be referenced.
Ans. All figure has been cited to the text.

Round 2
Reviewer 2 Report
Dear authors
The answer to self-citation is not satisfactory to me.
If you change all self-citation, I can accept your article to publish.
The article needs editing in English.
Author Response
Response to reviewer 2
The answer to self-citation is not satisfactory to me.
If you change all self-citation, I can accept your article to publish.
Ans. Dear reviewer, thank you very much for reviewing and accepting our work. Sure, I have removed the two following citations from the current research. Thank you very much.
- Ahmad, I.; Zhou, G.; Zhu, G.; Ahmad, Z.; Song, X.; Hao, G.; Jamal, Y.; Ibrahim, M.E.H. Response of leaf characteristics of BT cotton plants to ratio of nitrogen, phosphorus, and potassium. Pak. J. Bot 2021, 53, 873-881.
- Ahmad, I.; Zhou, G.; Zhu, G.; Ahmad, Z.; Song, X.; Jamal, Y.; Ibrahim, M.E.H.; Nimir, N.E.A. Response of boll development to macronutrients application in different cotton genotypes. Agronomy 2019, 9, 322.

Reviewer 3 Report
Review 2
Manuscript ID: ijms-2372430
“Phyto-Melatonin Role in Plant Growth and Physiology Under Abiotic
Stress”
Irshad Ahmad, Guanglong Zhu, Guisheng Zhou *, Jiao Liu, Muhammad
Usama Younas, Zhuyi Ming
The article has been greatly improved. The authors made good additions on page 2 lines 46-49 and clarified many phrases in the text. Figure 3 has been improved.
The reviewer had comments on the content of the Introduction, the design of Figures 2 and 4, and the renaming of paragraph 3.3.
The authors did not change the content of the Introduction. However, it does not make sense to describe one mechanism of action of melatonin in detail in the Introduction, while section 3 presents the other mechanisms of action of melatonin. In my opinion, in the Introduction it is necessary to present general statements about the known facts about the action of melatonin on the vital activity of plants. At the same time, references to articles with studies on NO2-stress can be given.
Page 4 lines 130-131.
Figure 2 is improved, but examples of stress types (“Salt stress”, “Drought stress”) should be attached with arrows to the “Abiotic Stress” figure. Melatonin should be taken out of the general chain of processes not flowing one from the other. Plant species (circle or ...) should also be shown with the same numbers.
Page 7 lines 228-229
Figure 4 is improved, but the diagram ends with Ca2+. Logically, melatonin changes Ca2+ levels and then stomatal movement. Ca2+ and the stomatal movement process should be rearranged (the stomatal movement process is not the cause of the change in Ca2+ levels). If necessary, an additional arrow with a "?" sign can be introduced from Ca2+.
The title of paragraph 3.3 should have been left as is.
NO2-stress should be introduced as a separate paragraph in section 3.

Author Response
Response to Reviewer 3
The article has been greatly improved. The authors made good additions on page 2 lines 46-49 and clarified many phrases in the text. Figure 3 has been improved.
Ans. Thank you very much for improving our work.
The reviewer had comments on the content of the Introduction, the design of Figures 2 and 4, and the renaming of paragraph 3.3.
The authors did not change the content of the Introduction. However, it does not make sense to describe one mechanism of action of melatonin in detail in the Introduction, while section 3 presents the other mechanisms of action of melatonin. In my opinion, in the Introduction it is necessary to present general statements about the known facts about the action of melatonin on the vital activity of plants. At the same time, references to articles with studies on NO2-stress can be given.
Ans. Dear reviewer, thank you very much. The introduction has been changed accordingly. The general statements about the known fact about the action of phto-melatonin on the vital activity in plants under abiotic stress have been discussed. In addition, according to your suggestion, The NO2 stress portion has been shifted to section 3.
Page 4 lines 130-131.
Figure 2 is improved, but examples of stress types (“Salt stress”, “Drought stress”) should be attached with arrows to the “Abiotic Stress” figure. Melatonin should be taken out of the general chain of processes not flowing one from the other. Plant species (circle or ...) should also be shown with the same numbers.
Ans. Figure 2 has been again changed accordingly.
Page 7 lines 228-229
Figure 4 is improved, but the diagram ends with Ca2+. Logically, melatonin changes Ca2+ levels and then stomatal movement. Ca2+ and the stomatal movement process should be rearranged (the stomatal movement process is not the cause of the change in Ca2+ levels). If necessary, an additional arrow with a "?" sign can be introduced from Ca2+.
Ans. The figure 4 has been changed and cleared now.
The title of paragraph 3.3 should have been left as is.
Ans. The title of the paragraph has been added.
NO2-stress should be introduced as a separate paragraph in section 3.
Ans. NO2 stress has been separately introduced in section 3.
Dear Reviewer once again thank you so much for improving our work.
